

# BioWorkbench: a high-performance framework for managing and analyzing bioinformatics experiments

Maria Luiza Mondelli[1], Thiago Magalhães[1], Guilherme Loss[1], Michael Wilde[2], Ian Foster[2], Marta Mattoso[3], Daniel Katz[4], Helio Barbosa[1,5], Ana Tereza R. de Vasconcelos[1], Kary Ocaña[1] and Luiz M.R. Gadelha Jr[1]

[1] National Laboratory for Scientific Computing, Petrópolis, Rio de Janeiro, Brazil
[2] Computation Institute, Argonne National Laboratory/University of Chicago, Chicago, IL, USA
[3] Computer and Systems Engineering Program, COPPE, Federal University of Rio de Janeiro, Rio de Janeiro, Rio de Janeiro, Brazil
[4] National Center for Supercomputing Applications, University of Illinois, Urbana, IL, USA
[5] Federal University of Juiz de Fora, Juiz de Fora, Minas Gerais, Brazil

Corresponding author
Luiz M.R. Gadelha Jr,
lgadelha@lncc.br

## ABSTRACT

Advances in sequencing techniques have led to exponential growth in biological data, demanding the development of large-scale bioinformatics experiments. Because these experiments are computation- and data-intensive, they require high-performance computing techniques and can benefit from specialized technologies such as Scientific Workflow Management Systems and databases. In this work, we present BioWorkbench, a framework for managing and analyzing bioinformatics experiments. This framework automatically collects provenance data, including both performance data from workflow execution and data from the scientific domain of the workflow application. Provenance data can be analyzed through a web application that abstracts a set of queries to the provenance database, simplifying access to provenance information. We evaluate BioWorkbench using three case studies: SwiftPhylo, a phylogenetic tree assembly workflow; SwiftGECKO, a comparative genomics workflow; and RASflow, a RASopathy analysis workflow. We analyze each workflow from both computational and scientific domain perspectives, by using queries to a provenance and annotation database. Some of these queries are available as a pre-built feature of the BioWorkbench web application. Through the provenance data, we show that the framework is scalable and achieves high-performance, reducing up to 98% of the case studies execution time. We also show how the application of machine learning techniques can enrich the analysis process.

## INTRODUCTION

Genome sequencing methodologies have led to a significant increase in the amount of data to be processed and analyzed by Bioinformatics experiments. Consequently, this led to an increase in the demand for their scalable execution. Most bioinformatics studies

aim to extract information from DNA sequences and can be classified as in silico experiments. In silico comprise mathematical and computational models that simulate real-world situations; they depend on the use of computational resources and specialized technologies for their execution. Simulations often require the composition of several applications, or activities, which have dependencies and manipulate large amounts of data. These aspects make it difficult to manage and control in silico experiments. Due to the complexity of simulation-based scientific experiments, it is necessary to use approaches that support their design, execution, and analysis, such as scientific workflows (*Deelman et al., 2009*). A scientific workflow is an abstraction that formalizes the composition of several activities through data set production and consumption. Each activity corresponds to a computational application, and the dependencies between them represent the execution data flow, in which the output of one activity is input to another.

Scientific Workflow Management Systems (SWfMS) can be used to manage the various steps of the life-cycle of a scientific workflow, i.e., design, execution, and analysis. SWfMS can be deployed in high-throughput computing (HTC) environments, such as clusters, clouds, or computational grids. They can also capture and store provenance information. Provenance records the process of deriving data from the execution of a workflow. For this reason, provenance describes the history of a given experiment, ensuring its reliability, reproducibility, and reuse (*Freire et al., 2008*). Provenance may also contain computational and scientific domain data, making it an important resource for the analysis of the computational behavior of an experiment and its scientific results. By combining computational and domain data, it is possible, for example, to make optimizations in the experiment as well as predictions for execution time and the amount of storage that it will require.

In this work, we present BioWorkbench: a framework that couples scientific workflow management and provenance data analytics for managing bioinformatics experiments in high-performance computing (HPC) environments. BioWorkbench integrates a set of tools that cover the process of modeling a bioinformatics experiment, including a provenance data analytics interface, transparently to the user. For managing scientific workflows, we use SWfMS Swift (*Wilde et al., 2011*), because of the support that the system provides for executing workflows in different HTC environments transparently and also due to its scalability (*Wilde et al., 2009*). Provenance data related to workflow performance and resulting data associated with the application area of the experiment are automatically collected by the framework. As part of the framework, we have developed a web application where workflow results are presented in a data analytics interface, from the abstraction of a set of queries to the provenance database, supporting the process of analysis by the scientists. We also demonstrate that it is possible to use machine learning techniques to extract relations from provenance data that are not readily detected, as well as to predict and classify the execution time of workflows. In this way, BioWorkbench consolidates various activities related to scientific workflow execution analysis in a single tool.

We evaluated the BioWorkbench using three case studies: SwiftPhylo, a phylogenetic tree assembly workflow; SwiftGECKO, a comparative genomics workflow; and RASflow,

a RASopathy analysis workflow. Each case study has its own characteristics. However, the three allowed the evaluation of aspects related to performance gains and provenance management covered by our framework. We show results where we obtained a reduction of up to 98.9% in the total execution time, in the case of SwiftPhylo, decreasing from ~13.35 h to ~8 min. Also, we demonstrate that with the provenance collected through the framework, we can provide useful results to the user who does not need to inspect files manually to extract this information.

## RELATED WORK

In general, our proposal addresses aspects related to the modeling of bioinformatics workflows, their parallel execution in HPC environments, as well as provenance data analytics, including predictions on computational resource usage. Here, we compare BioWorkbench to related solutions from these different points of view.

There are a variety of SWfMS for modeling and executing workflows in different application areas. Some of them allow execution in HPC environments. Among them, we highlight Pegasus (*Deelman et al., 2015*), which enables the specification of workflows through the XML format. The provenance is managed and recorded through the Wings/Pegasus framework (*Kim et al., 2008*) using the Ontology Web Language from W3C, and contains application-level and execution-level data that can be queried using the SPARQL language. Askalon (*Nadeem et al., 2007*) allows the definition of the workflow through a graphical interface, using Unified Modeling Language (UML) diagrams or the XML format. Workflow performance monitoring information is used by a service that supports planning the execution of activities. Taverna (*Wolstencroft et al., 2013*) is a SWfMS widely used by the bioinformatics community, where workflow activities usually comprise web services. Taverna has implicit iterations and parallelism, and its workflows can be further optimized to simplify complex parts using the DistillFlow optimization (*Cohen-Boulakia et al., 2014*). Provenance is collected and stored in a database and can also be exposed as W3C PROV in a portable Research Object (*Belhajjame et al., 2015*). Also, workflows can be shared through the *myExperiment* (*Goble et al., 2010*) platform. With a focus on cloud environments, SciCumulus (*Silva, Oliveira & Mattoso, 2014*) is a SWfMS based on relational algebra for workflow definition that uses a provenance database, which follows the PROV recommendation model from W3C, for configuration and monitoring of executions. It is distinct in that it allows provenance queries to be made during the execution of the experiment. Nextflow (*Di Tommaso et al., 2017*) is a framework comprised by a domain-specific language for the development of workflows. It is based on the dataflow programming paradigm and also implements an implicit parallelism model to execute its processes. Nextflow allows the user to enable the provenance record, or trace report, at the moment of the workflow execution and there is currently an effort to standardize this information to Research Object and W3C PROV. Compared with these solutions, we use the Swift SWfMS because, in addition to transparently supporting the execution of workflows in different HPC environments, it has been shown that it has a high potential for scalability (*Wilde et al., 2009*), and it supports provenance management (*Gadelha et al., 2012*). Also, it supports workflow

patterns such as conditional execution and iterations that not supported by similar systems such as Pegasus (*Deelman et al., 2015*). It also evaluates the workflow dynamically, possibly changing task execution sites for efficiency of reliability.

*Juve et al. (2013)* present Wfprof, a tool for collecting and summarizing performance metrics of workflow activities. Wfprof was coupled to the Pegasus SWfMS to identify the complexity level of different workflows and how computational and I/O intensive they are. In *Król et al. (2016)*, an approach for analyzing the performance of workflows executed with the Pegasus SWfMS is presented, which also studies the effect of input parameters on task performance. The PDB (*Liew et al., 2011*) presents an approach that collects and stores computational data in a database for planning the execution of workflows that occur in memory and out-of-core. ParaTrac (*Dun, Taura & Yonezawa, 2010*) is a data-intensive workflow profiler with an interface that allows its use by different SWfMS. ParaTrac uses the Linux *taskstats* interface to collect memory usage, CPU and runtime statistics, and the FUSE file system to record the amount of data that is passed between workflow activities. In *Silva et al. (2016)*, provenance data is integrated into the TAU code profiling tool, allowing the performance visualization of workflows executed with the Scicumulus SWfMS. This work aims to carry out the monitoring and profiling for the detection of anomalies in the execution of large-scale scientific workflows. Visionary (*Oliveira et al., 2017*) is a framework for analysis and visualization of provenance data with the support of ontology and complex networks techniques. However, the work is focused on the analysis of the provenance graph and does not include domain data or predictive analysis. Our approach provides profiling analysis of the execution of workflows from the provenance collected by Swift, in addition to being able to aggregate domain data from the experiment. By combining these two types of information we can add even more value and gain insights from the process of analyzing the results of the experiment.

A survey of existing frameworks and tools for bioinformatics workflows is presented in *Leipzig (2017)* and *Fjukstad & Bongo (2017)*. Both highlight the need to enable workflow reproducibility and scalability, aspects that we approach with the development of BioWorkbench. Also focusing in bioinformatics workflows, the WEP (*D'Antonio et al., 2013*) tool enables the modularization of the workflow, allowing the user to execute the entire experiment or just the necessary activities. In addition, WEP also allows access to intermediate files and final results through the web interface. It does not provide detailed computational profiling or parallel and distributed execution features as does BioWorkbench. ADAM (*Massie et al., 2013*) is a scalable API for genome processing workflows that uses Spark (*Zaharia et al., 2016*), a framework for managing Big data. Both tools allow for running experiments in HPC environments, but they implement parallelism at a lower level, within the activities. ADAM requires the API functions to be implemented according to Spark's model of computation, using distributed datasets and applying actions and transformations, such as map and reduce, to these. The bioKepler (*Altintas et al., 2012*) is an approach to facilitate the development and execution of workflows in distributed environments using the Kepler SWfMS. Like BioWorkbench, bioKepler is motivated by the challenges brought by the advancement of sequencing techniques. However, the tool uses data-parallel execution patterns that require greater

attention from users when defining the workflow. BioWorkbench allows for the parallel composition of workflow activities that may be implemented using different models of computation. Therefore, BioWorkbench is more flexible in incorporating existing and legacy bioinformatics tools into its workflows.

Related to the reproducibility of workflows, (*Kanwal et al., 2017*) highlights the importance of having sufficient provenance information to support the understanding of data processing and ensuring the consistency of the workflow results. To demonstrate the challenges that arise in the process of reproducing the results, a variant calling workflow was implemented using three different SWfMS. They show a set of aspects associated with each SWfMS that hinder the understanding and reproducibility of workflows, including lack of documentation and provenance data. For each aspect, the authors propose recommendations that, together with provenance patterns, could facilitate reproducibility. Our approach supports some recommendations raised by the work, such as the availability of workflows and the framework through public repositories. In addition, through machine learning techniques, we demonstrate that it is also possible to predict the execution time of workflows based on provenance data from previous executions. In future work, the use of these techniques may support other recommendations raised by the authors.

## MATERIALS AND METHODS

### Design and implementation

BioWorkbench (https://github.com/mmondelli/bioworkbench; *Mondelli, 2018a*) aims to support users in the specification process of an experiment, the experiment execution in HPC resources, the management of consumed and produced data, and the analysis of its execution results. BioWorkbench uses a SWfMS for definition and execution of bioinformatics workflows, and a web application for provenance data analytics. Figure 1 shows the layered conceptual architecture of BioWorkbench. It should be noted that this architecture is available for demonstration in a Docker (*Boettiger, 2015*) software container (https://hub.docker.com/r/malumondelli/bioworkbench/) that allows one to reproduce the computational environment described and used in this work. In this way, BioWorkbench supports reproducibility at two levels: the computational environment and the data derivation history (through provenance information). Docker also supports better maintenance, integration, and distribution of the BioWorkbench framework and its components. All the software components are clearly defined in a container specification file, which describes their dependencies and how they are interconnected. The development process of BioWorkbench is integrated with its respective container, every modification in BioWorkbench triggers the construction of a new container that allows for rapidly testing new versions of the framework. In the following subsections, we detail the features and functionalities of each layer.

### Specification and execution layer

This layer uses Swift (*Wilde et al., 2011*), a SWfMS that allows users to specify workflows through a high-level scripting language and execute them in different HPC

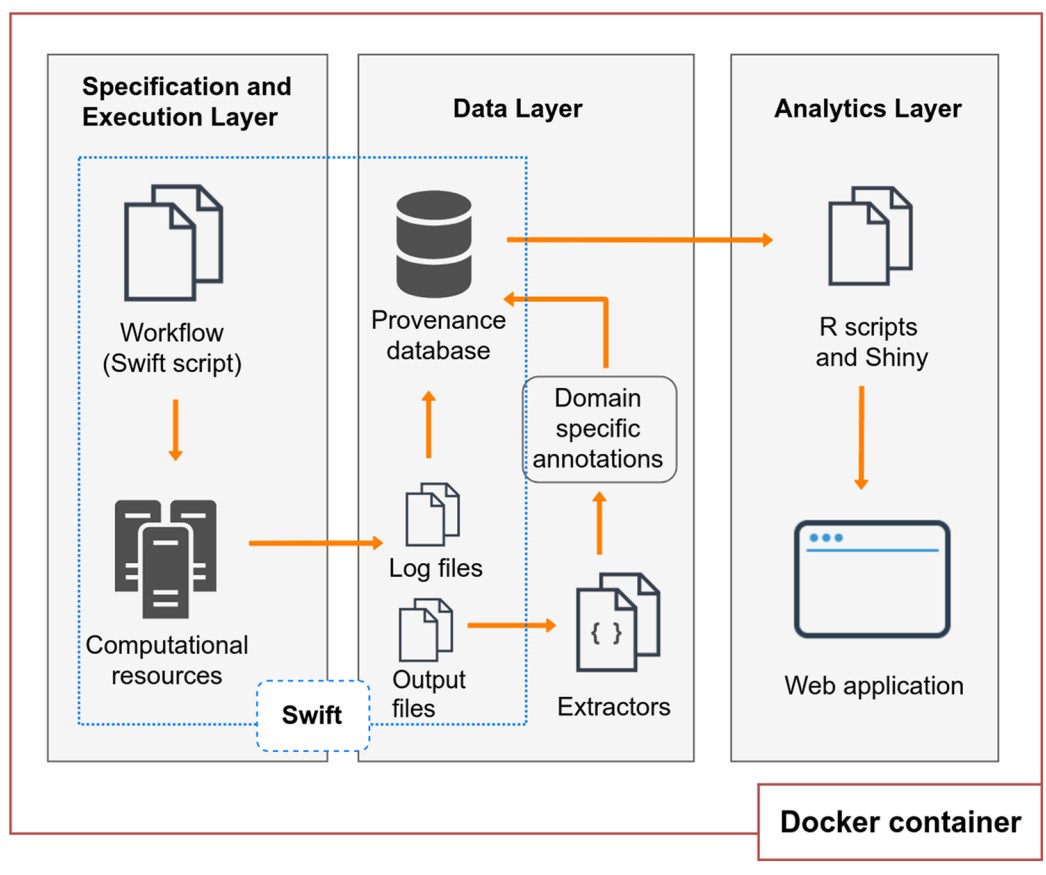

**Figure 1 BioWorkbench layered conceptual architecture.**

environments. The scripting language follows functional programming principles that allow, for example, all operations to have a well-defined set of inputs and outputs and all the variables to be write-once. Through the Swift scripting language, datasets are declared as variables of primitive types, such as floats, integers, and strings; mapped types, which associate a variable with persistent files; and collections, which include arrays and structures. The activities that comprise a workflow are represented as *app functions*, responsible for specifying how applications outside Swift should be executed, also indicating their input and output files or parameters. *Compound functions* are used to form more complex data flows and can include calls to *app functions*, loops, and conditionals. The activity chaining is defined in the script, by indicating that the output of an *app function* is an input to another *app function*. An example of activity chaining is presented in Listing 1. In Swift all expressions are evaluated in parallel as soon as their data dependencies are met. Through the *foreach* loop instruction, it is possible to process all elements of an array in parallel. The independence of locality is another essential feature of Swift and allows the same workflow to run on different computing resources without the need to change its specification. Therefore, Swift supports the management and parallel execution of these workflows in HPC environments. This layer is then responsible for managing, through Swift, the execution of the workflow in the computational resources to which the user has access and intends to use.

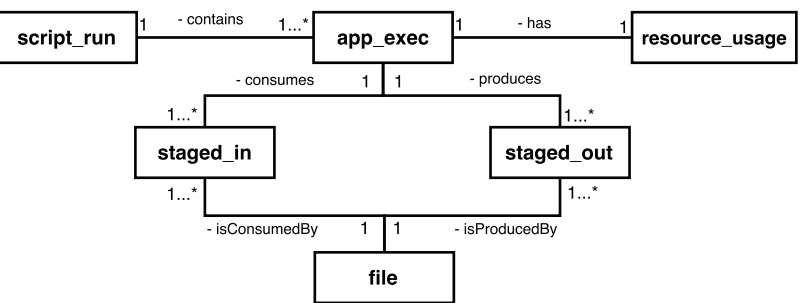

**Figure 2 Main entities in the conceptual model of the Swift provenance database.**

## Data layer

The data layer handles the provenance of workflow executions. For this, the layer also uses Swift because of its capability of tracking provenance of the workflow execution behavior as well as its activities (*Gadelha et al., 2011*). All information regarding the execution of a workflow in Swift, or its provenance, is recorded in a log file. Each of the activities executed by a workflow also has a log file, called *wrapper log*. These files contain information such as the computational aspects of activity executions: the number of read and write operations to the file system, CPU and memory utilization, and other details. Also, the log files keep track of the files consumed and produced by each of the activities, analyzing the data derivation process of the experiment.

To provide access to provenance, Swift has a mechanism that processes these log files, filters the entries that contain provenance data, and exports this information to a relational SQL database (SQLite) (*Mondelli et al., 2016*). The main entities of the Swift provenance model are presented in Fig. 2. The *script_run* entity contains the description of the execution of a workflow; the *app_exec* entity describes aspects of the activities associated with a workflow, and *resource_usage* has the computational information of the executions of those activities. The *file* entity has the record of all the files consumed and generated by the execution of a workflow; *staged_in* and *staged_out* entities, in turn, record the process of deriving data, relating the files to the executions of the activities.

Workflow executions may also produce domain provenance data related to the application area of the experiment. Domain data comprises the results obtained by the execution of computational activities of the workflow or parameters used by them. Usually, these results are stored in files. In bioinformatics, for example, results from a sequence alignment experiment may consist of rates or scores, indicating to the scientist the accuracy of the alignment. Thus, domain data is data essential for bioinformatics analyses. In order to store this type of information along with the provenance Swift collects by default, the *Data Layer* can use a set of extractors developed for collecting domain data, or annotations, from some bioinformatics applications. Extractors consist of scripts that collect domain data from the files produced by the activities, and that can support the analysis process by the user. These annotations can be exported to the provenance database and associated with the respective workflow or particular activity execution in the form of key-value pairs.
The provenance database contains a set of annotation entities, which relate to the model presented, giving more flexibility to store information that is not explicitly defined in the schema. More entities can be added to the database allowing for better management of the results of the workflow execution, depending on the type of annotation to be stored. In the Results and Discussion section, we describe some of the entities that have been added to the Swift provenance model in order to provide better domain data management of a case study. Once the provenance data is stored in the database, they can be accessed by the *Analytics Layer*, described in the next subsection, in order to facilitate access and visualization of the results. Also, we highlight that it is possible to export the Swift provenance database to the Open Provenance Model (OPM) model and we provide a script for this purpose in the Docker container. In a future work, one possibility would be to upgrade this export to the PROV model.

## Analytics layer

This layer abstracts a set of database queries from the data layer, facilitating the access to the provenance of the workflow. The main goal is to provide the user with a more intuitive way to understand the computational profile of the experiment and to analyze the domain results. Without the set of query abstractions that we propose in this layer, the user would have to manipulate both the provenance database and the output files of the experiment. As part of the query abstraction process, we use R scripts that connect to the provenance database, perform the queries and extract useful statistics for analysis. The results of the queries retrieve data that are presented to the user through graphs and tables in a web interface. This interface was developed using the Shiny library (https://shiny.rstudio.com/), which allows an analysis implemented in R to be accessed via web applications interactively. The interface provides a menu so that researchers can carry out their analyses. This menu, shown in Fig. 3, allows them to select a workflow execution from the list of executions available for analysis. The charts and tables are updated according to the chosen options. Computational analyses present graphs such as the execution time of the activities, the percentage of CPU used by each of them, and the parallelism level of the workflow. Information about the total time of execution of a workflow, the number of executed activities, and whether the workflow ended successfully is also displayed. Domain analyses provide information about the scientific domain of the workflow. In the RASflow case study, presented in the Results section, this information includes data on the genetic mutations found in patients.

## Machine learning techniques in support of provenance analysis

Provenance data can be a useful resource for a wide variety of machine learning algorithms for data mining. The application of machine learning algorithms in this context consists of a set of statistical analyses and heuristic algorithms that enable the automatic generation of models aiming to describe some data of interest. These models can produce: (i) predictions, allowing scientists to estimate results or behavior based on some previous information; and (ii) learning, related to the discovery of implicit relations that are not always detected by human observations or simple queries.

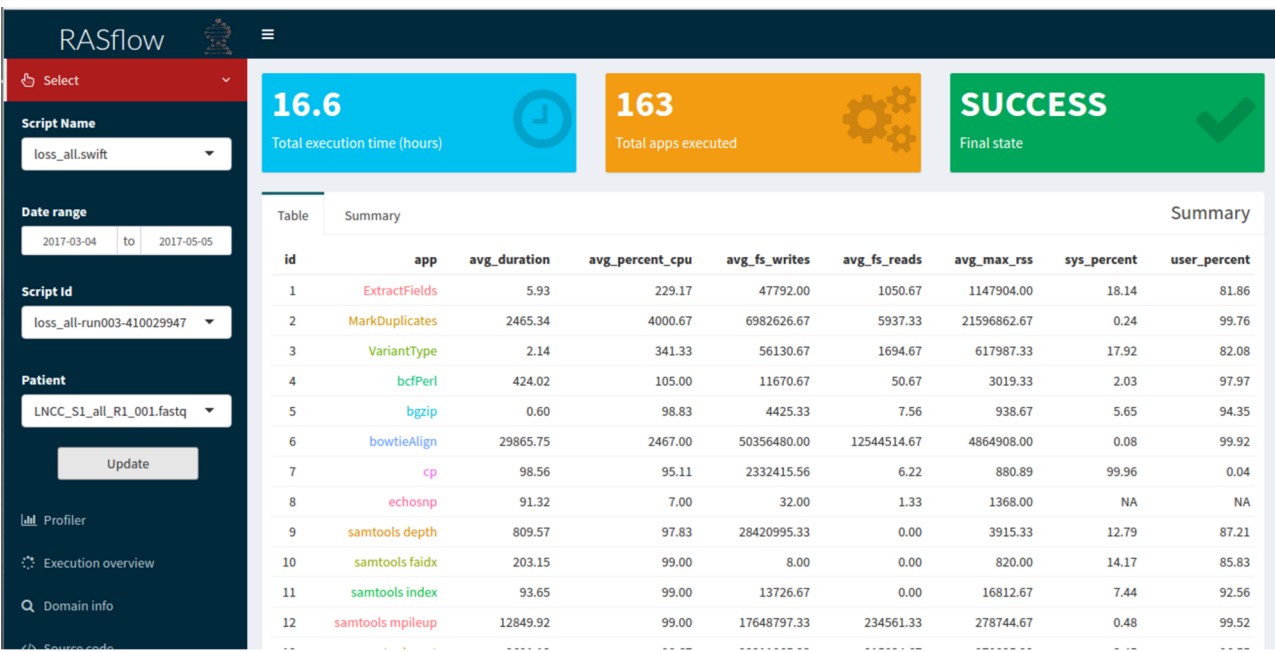

**Figure 3** BioWorkbench web interface displaying information about a RASopathy analysis workflow (RASflow) execution.

From a data set of interest, the models take into account a set of attributes that must be passed as input to produce an output. If the output can assume values in a continuous numeric range there is a regression problem and the models are equations that assign real values to the received instances. However, if the outputs can only assume a previously defined set of discrete values, there is a classification problem and the models map the input attributes to classes defined by these discrete values. In bioinformatics workflows, for example, we can have as input to a machine learning algorithm a set of attributes such as genome sequence size, statistics on memory usage and processing demand. The output of this algorithm may be the prediction of the execution time. In comparative genomics, more specifically, behaviors or relations such as the required computational time, memory or space for comparing an arbitrary number of genomes can also be predicted.

In this work we combine provenance data and machine learning by using the free software Weka (*Hall et al., 2009*) (version 3.6) to provide a preliminary analysis that endorses the usability of this methodology to study workflows behavior. Weka provides a broad set of machine learning and statistical analysis techniques for data mining that include rule- and tree-based algorithms, such as *J48*, *BFTree*, *OneR*. We devised BioWorkbench to be a provenance analytics web application. We initially incorporated simple queries in the web interface that could provide some statistics on workflow executions and also abstractions for more complicated (non-predictive) queries. However, we intend to add more intelligent predictive analysis to the web interface. The analyses described in this section, implemented as scripts in Weka, are a first step in this direction. They take data directly from the provenance database and perform

predictions on computational resource usage. A next step will be to incorporate the functionality provided by these scripts to the web interface of BioWorkbench. Therefore, we view the machine learning scripts as a *command-line* component of our framework. Also, we highlight that the commands used for the machine learning analyses presented in this work are available in the BioWorkbench repository and its Docker container.

Thus, we hope to indicate some relevant opportunities that arise from the association between provenance data and machine learning techniques, both in the specific case of SwiftGECKO, presented below, and in the scientific workflows field. Also, these experiments can contribute to the understanding of workflow behavior, being useful, for instance, in guiding optimization efforts and parameter configurations. It is worth mentioning that the techniques used in this work were applied using the original algorithm attributes proposed in the software Weka, except for the minimum number of tuples classified at a time by the models. This attribute influences the model complexity, providing a minimum amount of instances that need to be simultaneously classified by the model for each possible input parameters configuration. The values of this parameter were changed to reduce the complexity of the models and consequently to increase their understandability. A more detailed description of the algorithms included in Weka, together with some statistical information that we did not include in this work for scope reasons, can be found in papers such as (*Arora & Suman, 2012*) and (*Sharma & Jain, 2013*).

The RASflow workflow, described in the next section, was used in another study on RASopathies. In this article, we only demonstrate the functionality of the bioinformatics tool used in that study. The associated research project was examined and approved by the Fernandes Figueira Institute Ethics Committee at Oswaldo Cruz Foundation, document number CAAE 52675616.0.0000.5269. All participants signed an informed consent before enrollment in the study.

## RESULTS AND DISCUSSION

We present three case studies to evaluate BioWorkbench: SwiftPhylo, a phylogenetic tree assembly workflow; SwiftGECKO, a comparative genomics workflow; and RASflow, a RASopathy analysis workflow. These workflows are analyzed from both computational and scientific domain points of view using queries to a provenance and annotation database. Some of these queries are available as a pre-built feature of the BioWorkbench web interface. The values and statistics presented in this section were extracted from queries to the Swift provenance database. In the case of SwiftGECKO and RASflow, in addition to the computational information already collected and stored by Swift, annotations were also gathered from the scientific domain of the experiment, such as the size of the sequences used as input for the workflow execution. With this, we show that it is possible to combine provenance and domain data for more detailed analyses. Also, in the SwiftGECKO case, we demonstrate that these analyses can benefit from machine learning techniques for extracting relevant information and predicting computational resource consumption that are not readily detected by queries and statistics from the provenance database. These aspects demonstrate the usefulness of

Swift both in supporting the parallel execution of the experiment and in the analysis of those runs through queries to the provenance database. The executions of each of the workflows were performed on a shared memory computer, with a single node with 160 cores and 2 TB of RAM. It is worth mentioning that the computational resource is shared with other users and was not dedicated to these executions. This can be considered as one of the factors influencing the performance gains. Also, the computational results are related to the computational resource that we use to execute the experiments. The workflows can be replicated on other resources in which users have access to, however, computational results may vary. We were able to reserve part of the machine for the executions, we had exclusive access to a number of CPUs that corresponded to the number of threads used in each benchmark. To try to dissipate I/O effects on the benchmarks, due to processes running in the other CPUs, we performed multiple executions for each benchmark. We highlight that the workflows were executed directly in the aforementioned computational resource, not taking into account the Docker structure presented in the Fig. 1. The Docker container was built for reuse purposes only, to encapsulate the components that compose the framework.

## SwiftPhylo: phylogenetic tree assembly

SwiftPhylo (https://github.com/mmondelli/swift-phylo; *Mondelli, 2018d*) is based on the approach proposed in *Ocaña et al. (2011)*. Its goal is to build phylogenetic trees from a set of multi-FASTA files with biological sequences from various organisms. The construction of phylogenetic trees allows to study the evolutionary relationship between organisms, determining the ancestral relationships between known species. Results obtained through phylogenetic experiments contribute, for example, to the development of new drugs (*Anderson, 2003*). SwiftPhylo is composed of six activities, shown in Fig. 4 and described as follows:

1. Sequence numbering: the activity receives multi-FASTA format files as input and uses a Perl script so that each sequence contained in the file receives is numbered.
2. Multiple sequence alignment (MSA): the activity receives the result of Activity 1 and produces a MSA as output through the MAFFT (version 7.271) (*Katoh et al., 2002*) application.
3. Alignment format conversion: this activity converts the format of the file generated by the previous activity to the PHYLIP format using the ReadSeq (version 2.1.30) (*Gilbert, 2003*) application.
4. Search for the evolutionary model: this activity tests the output files of Activity 3 to find the best evolutionary model, through the ModelGenerator (version 0.84) (*Keane et al., 2006*) application.
5. Filtering: this activity uses a Python script to filter the output file from Activity 4.
6. Construction of phylogenetic trees: this activity receives as input the resulting files of Activities 3 and 5, and uses the RAxML (version 8.2.4) (*Stamatakis, 2006*) application to construct phylogenetic trees.

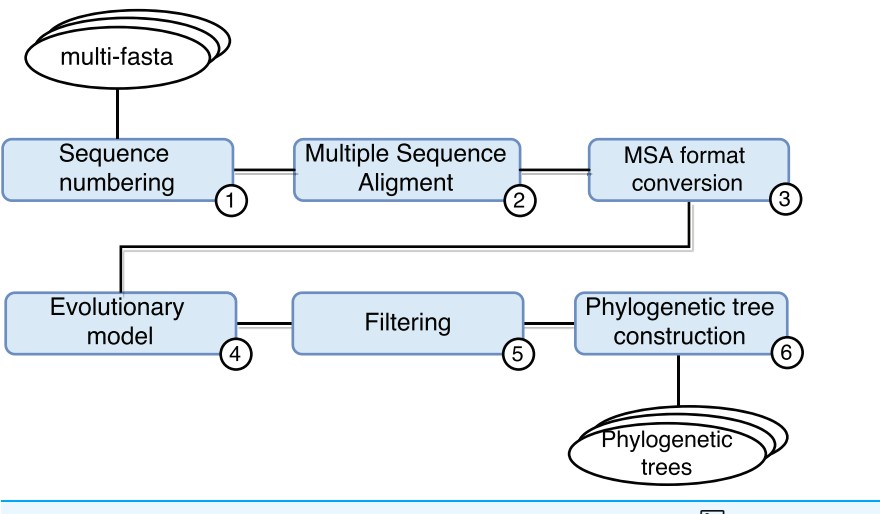

**Figure 4 SwiftPhylo workflow modeling.**

SwiftPhylo works with a large amount of data and can be run with different parameters. This means that, in practice, managing its execution without the help of a SWfMS becomes a cumbersome task for the scientist. Taking into account that the workflow modeling specifies independent processing for each input file, with the implementation of SwiftPhylo we have a workflow that allows us to explore the characteristics of Swift's parallelism.

In the SwiftPhylo implementation process, the computational applications that compose the workflow were mapped to the Swift data flow model. In this way, the activities are represented in the Swift script as *app functions*. The app functions determine how the execution of a computational activity external to Swift is performed, including its arguments, parameters, inputs, and outputs. Once this has been done, activity chaining was defined by indicating that the output of one app function is the input of another. Parallelism has been implemented in a way that the activity flow can be executed independently for each input, as shown in the code in Listing 1 through the *foreach* statement.

**Listing 1 SwiftPhylo specification sample.**

```
app (file o) mafft (file i) {
  mafft filename(i) stdout=filename(o);
}
foreach f, i in fastaFile {
  mafftFile[i] = mafft(fastaFile[i]);
}
```

SwiftPhylo was executed using a set of 200 multi-FASTA files, with the size of each file ranging from 2 to 6 KB, resulting in the execution of 1,200 activities. The total execution time averages and speedup of the workflow are shown in Fig. 5. The difference between the slowest and the fastest execution for each number of cores varies between 0.04% (1 core) and 3.38% (160 cores) or the total execution time, which are too small to represent in figure as error bars. The results show that SwiftPhylo executed in parallel was ~92 times faster than its sequential execution, i.e., a ~13.35 h

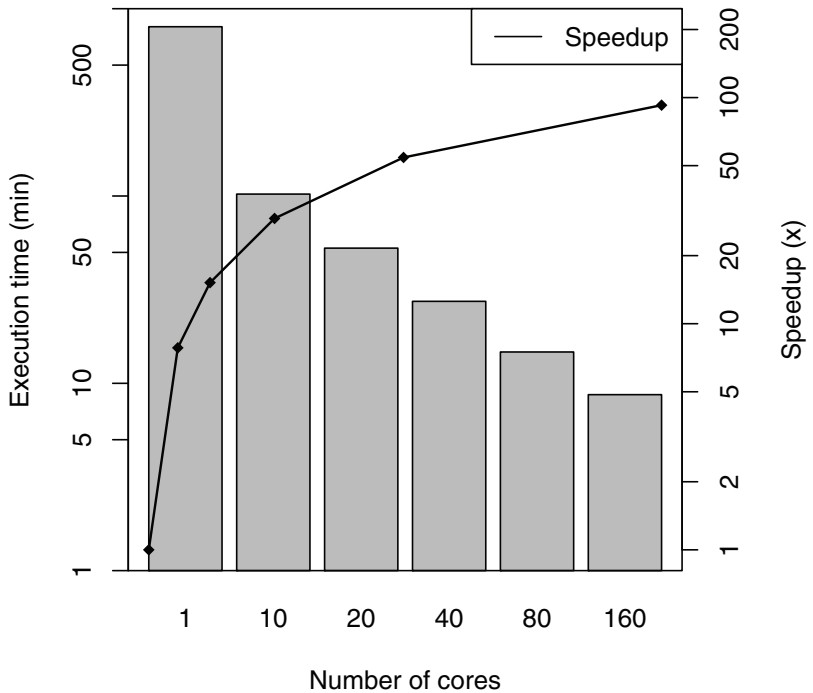

**Figure 5 SwiftPhylo execution time and speedup.**

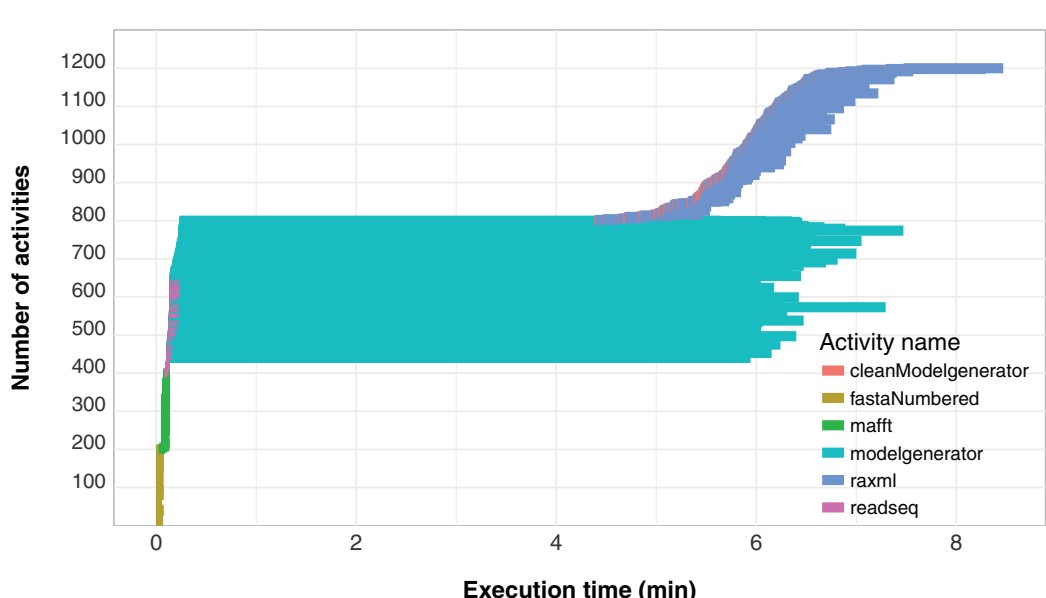

**Figure 6 SwiftPhylo workflow Gantt chart expressing its parallelism level.**

execution time was reduced to ~8 min. This represents a decrease of 98.9% in the workflow execution time.

Another important aspect related to the execution of the workflow concerns its level of parallelism, presented through a Gantt graph in Fig. 6. This type of analysis shows the order of execution and the duration of each of the activities that comprise the *workflow*.

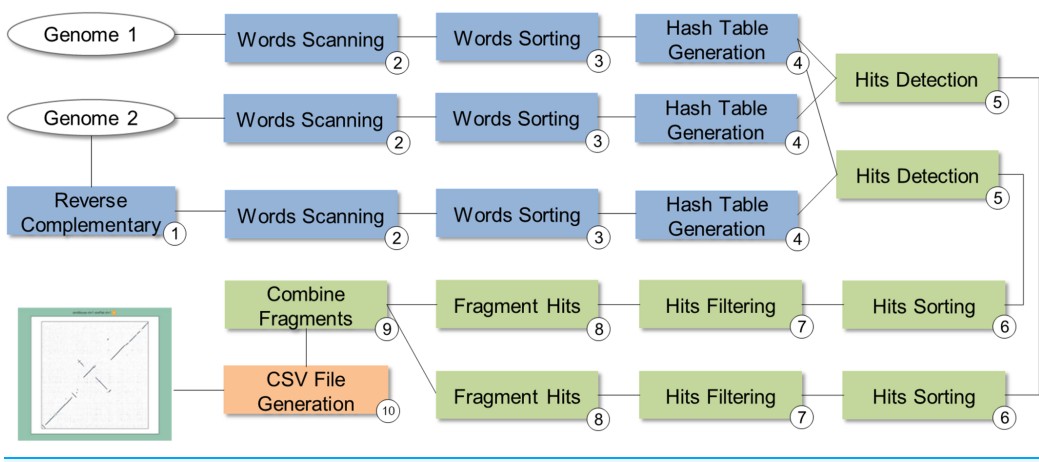

**Figure 7 SwiftGECKO workflow modeling.**

In the chart, activities are represented by different colors and each activity is displayed as a horizontal bar that indicates when the activity started and when it ended. The bars are stacked on the vertical axis in order of execution. Thus, by plotting a vertical line at some time $t$ of the workflow execution, we find the parallelism at time $t$ as the number of activities intercepted by the line. We can observe that the *modelgenerator* activity is the one that demands the most execution time and can be considered a candidate for identify parallelism strategies that reduce its duration.

## SwiftGECKO: comparative genomics

SwiftGECKO (https://github.com/mmondelli/swift-gecko; *Mondelli, 2018c*) is an implementation of the comparative genomics workflow proposed in *Torreno & Trelles (2015)* (version 1.2). Comparative genomics studies the relationship between genomes of different species, allowing a better understanding of the evolutionary process among them (*Alföldi & Lindblad-Toh, 2013*). SwiftGECKO aims to identify portions of biological sequences of various organisms with a high degree of similarity, or high-scoring segment pairs (HSP), between them. SwiftGECKO is composed by 10 activities distributed in three modules, which are presented in Fig. 7 and described as follows:

1. Dictionary creation: this corresponds to the creation of dictionaries for each sequence and includes Activities 1–4 (blue boxes). For the dictionary creation, the user must indicate the size of the portion of the sequence, or *K*-mers, that will be used for the comparison. The dictionary consists of a table that organizes the *K*-mers and their respective positions, for each of the sequences.

2. HSP identification: this is composed by Activities 5–9 (green boxes) and performs the comparison between the sequences, identifying *hits* used to find HSPs.
   *Hits* consist of positions where the *K*-mers of the compared sequences are equal.

3. Post-processing: this is Activity 10, where a conversion of the output format of Activity 9 to CSV is done, allowing analysis of the results. The CSV file contains information such as the size of the sequences, the parameters used in the comparison, and the number of hits found.

**Table 1 Average duration (s) and the amount of data read and written by each activity of SwiftGECKO.**

| Activity | Duration (s) | GB read | GB written |
|---|---|---|---|
| hits | 60,058 | 455.24 | 111.36 |
| sortHits | 4457.3 | 111.36 | 111.36 |
| FragHits | 3793.3 | 94.7 | 0.32 |
| filterHits | 2,402 | 111.36 | 83.09 |
| csvGenerator | 697.8 | 0.006 | 0.004 |
| combineFrags | 425.8 | 0.32 | 0.16 |
| w2hd | 425.1 | 7.05 | 11.67 |
| sortWords | 76.6 | 7.05 | 7.05 |
| words | 55.9 | 0.29 | 7.05 |
| reverseComplement | 11.8 | 0.15 | 0.14 |

SwiftGECKO consumes and produces a significant amount of data. A workflow execution with 40 complete bacterial genomes totals 1,560 possible comparisons, resulting in the execution of 8,080 activities. The implementation of SwiftGECKO followed the same steps mentioned in the SwiftPhylo implementation process. Parallelism was divided into two *foreach* stages as shown in Listing 2. In the first one, the activities that belong to Module 1 are executed; they generate the files that will be consumed by the other activities. The second step, referring to Modules 2 and 3, manages the activities responsible for comparing pairs of all genomes given as input.

**Listing 2 SwiftPhylo specification sample.**

```
foreach f,i in fasta {
    wordsUnsort[i] = words(fasta[i]);
}
foreach f,i in fasta {
    foreach g,j in fasta {
      hits[i][j] = hits(d2hW[i], d2hW[j], wl);
    }
}
```

SwiftGECKO was executed with a set of 40 files containing bacterial genome sequences, downloaded from NCBI (https://www.ncbi.nlm.nih.gov/), ranging in size from 1 KB to 8 MB. This set of genomes is available along with the workflow script in the GitHub repository. The execution time decreased from ~2.1 h to ~6 min, representing a reduction of ~96.6%. This means that, the workflow was ~30 times faster when executed in parallel, using 160 processors. In a more in-depth analysis, we illustrate in Table 1 that the *hits* activity is also I/O-intensive. This is a factor that limits the scalability of the workflow.

In addition, as aforementioned, we present predictive models automatically generated by machine learning methods. We choose the CPU time as the attribute to be predicted, thus the produced models aim at understanding how some variables influence the total execution time of the workflow. Besides the CPU time and the total_fasta_size, we included the following attributes as input parameters for the models: *length*, *word_length*,

*similarity*, *total_genomes*, *total_reads*, and *total_written*. The workflows were executed three times with different parameter configurations, and their respective execution times were recorded in a database containing 156 instances that were used as input to the machine learning algorithms. Thus, to evaluate the performance of each generated model we used metrics related to the differences between the predicted and the previously known CPU times, which were obtained through these previous executions. Whenever possible the algorithms were evaluated using the methodology known as "cross-validation," with 10-fold (*Arlot & Celisse, 2010*) and, because of this, all the measurements presented here with respect to the machine learning models also express their generalization potential.

Firstly, the regression algorithms obtained especially promising results. The simple linear regression algorithm achieved a correlation coefficient of 0.9856 and is represented in the Eq. (1). The greater the coefficient correlation, the more precise is the prediction of the algorithm. Some attributes are handled as nominal ones, being able to assume only a set of discrete values. For example, *word_length* can be set as 10, 12, or 16. For continuous attributes there is always a multiplicative constant associated with the current value of the attribute. However, for nominal cases, a set of possible values to the attribute are associated with the constant. If the attribute assumes one of these, then the constant is multiplied by 1. Otherwise, it is multiplied by 0. Thus, the linear regression model is an equation that receives a set of values as input parameters and returns a real number that aims to approximate the real execution time of a workflow represented according to these parameters. The multilayer neural network algorithm obtained a correlation coefficient equal to 0.9936 but we have not represented this due to its high complexity. These results demonstrate that it is quite possible to predict the execution time of the workflow as a whole, on the chosen set of input parameters. In other words, they indicate the existence of a relation between these parameters and the execution time. Also, *word_length*, *total_written*, and *total_genomes* are highlighted by the generated model as the most relevant parameters to predict the execution time of the workflow.

$$
\begin{aligned}
CPU\ time = \quad & -931.4025 & \times & \ (word\_length\ in\ \{10, 12\})+ \\
& -2704.5534 & \times & \ (word\_length = 10)+ \\
& 0 & \times & \ (total\_written)+ \\
& -1006.3049 & \times & \ (total\_genomes\ in\ \{20, 30, 40\})+ \\
& -547.6854 & \times & \ (total\_genomes\ in\ \{30, 40\})+ \\
& 410.1449 & \times & \ (total\_genomes = 40)+ \\
& 876.1037 &
\end{aligned}
\tag{1}
$$

We divided the domain of possible values for *CPU time* in five ranges (which are the classes) {*A, B, C, D, E*} of equal size, in which A includes the lowest values and E includes the highest values. This made it possible to apply classification algorithms using the same data as the previous analysis. Also, in order to produce predictive models, we excluded data that could provide information obtained after the workflow execution, and not before. Thus, we excluded the attribute "total_written." Having a discrete domain, the information gain ratio values for each attribute are listed in the Table 2. Greater

**Table 2 Information gain ratio values of the attributes used for the analysis.**

| Information gain ratio | Attribute |
| --- | --- |
| 1 | total_read |
| 0.1842 | word_length |
| 0.1611 | total_fasta_size |
| 0.1208 | total_genomes |
| 0.0421 | length |
| 0.0271 | similarity |

this value, greater is the amount of information that the respective attribute offer to predict the execution time of the workflow. Cost matrices were used to avoid the overvaluation of class A, which holds 88.46% of all tuples. In this case, the tree-based techniques "J48" (*Quinlan, 1993*) and "BFTree" (*Shi, 2007*) obtained the best results, correctly classifying 96.79% and 96.15%, respectively, of all tuples.

The "OneR" algorithm (*Holte, 1993*) chosen automatically the *total_read* attribute to produce a notably simple model, given by the Eq. (2) and that classified correctly 96.79% of all tuples.

$$
\begin{aligned}
total\_read : & \\
< 1.768799155948E12 \quad &: \quad Class\ A \\
< 2.5173879304975E12 \quad &: \quad Class\ B \\
< 3.4581148731605E12 \quad &: \quad Class\ C \\
\geq 3.4581148731605E12 \quad &: \quad Class\ D
\end{aligned}
\tag{2}
$$

According to the Table 2, the *CPU time* is most directly influenced by the amount of data read by the workflow. Together with the third higher information gain ratio value assigned to the attribute *total_fasta_size*, we can suggest that the magnitude of the information to be read is the most influential aspect concerning the execution time. Despite the I/O routines, the attribute *word_length* receives the second higher information gain ratio value, highlighting its relevance.

In fact, in the absence of data about the number of hits, the attribute *word_length* assumes the most prominent position to predict the attribute *CPU time*. For all this, the value of *word_length* is also a decisive attribute to predict the execution time of the workflow as a whole, due to its importance in the behavior of the most costly component of the workflow.

Reinforcing the relevance of the I/O routines, both Algorithms 1 and 2 employ the attributes *total_read* and *total_fasta_size* in their main conditionals. As in the Eq. (2), the greater the *total_read* value, the greater the computational time demanded. Also, the attributes *length* and *similarity* are used to classify a minor number of tuples. However, they can indicate the influence of the cost associated with the amount of fragments found (influenced by the attributes *length* and *similarity*).

Beyond all this, these experiments demonstrate that, having an appropriated set of input data, is possible to generate simple models able to efficiently predict the computational time demanded by the workflow execution, both in terms of continuous or discrete times.

**Algorithm 1** J48 predictive model.

1 **if** *total_read* $\leq$ *1020229446482* **then**
2    **return** Class A;
3 **else**
4    **if** *total_fasta_size* $\leq$ *113556442* **then**
5       **if** *length = 80* **then**
6          **if** *similarity = 65* **then**
7             **return** Class C;
8          **else**
9             **return** Class B;
10       **else**
11          **return** Class C;
12    **else**
13       **return** Class E;

**Algorithm 2** BFTree predictive model.

1 **if** *total_read* $\leq$ *1.768799155948E12* **then**
2    **return** Class A;
3 **else**
4    **if** *total_fasta_size* $\leq$ *1.314087315E8* **then**
5       **if** *length=100* **then**
6          **return** Class C;
7       **else**
8          **if** *similarity* $\neq$ *40* **then**
9             **return** Class C;
10          **else**
11             **return** Class B;
12    **else**
13       **return** Class E;

The techniques used to build these models also allows us to infer predictions related to other attributes, beyond the execution time. For example, we can focus on the amount of written data by the workflow or the specific domain data stored in the provenance database. Also, there are a wide variety of machine learning methods that produce symbolic solutions and allow, in addition to the predictions, knowledge extraction about different aspects. These aspects include, for example, the structure or the importance degree of relations among the various variables or between variables and constants. Therefore, machine learning methods constitute an essential toolkit to be explored in provenance data analyses concerning SwiftGECKO and other, to provide predictive models and to reveal implicit knowledge.

## RASflow: RASopathy analysis

Genetic diseases, such as RASopathies, occur due to changes in the nucleotide sequence of a particular part of the DNA. These changes modify the structure of a protein, which may cause anatomical and physiological anomalies (*Klug, Cummings & Spencer, 1997*). RASopathies comprise a set of syndromes characterized by heterogeneity of clinical signs and symptoms that limit the prognosis, still in the childhood, of predispositions to certain tumors or neurocognitive delays. RASopathies are characterized by mutations in genes that encode proteins of the RAS/MAPK cell signaling pathway, which is part of the communication system responsible for the coordination of cellular activities and functions. This type of mutation was found in 20–30% of cancer cases in humans, setting the RASopathies in the group of syndromes that are predisposed to cancer (*Lapunzina et al., 2014*).

A consortium between the Bioinformatics Laboratory of the National Laboratory for Scientific Computing (LABINFO/LNCC) and the Center for Medical Genetics of the Fernandes Figueiras Institute at Fiocruz (IFF/Fiocruz) established an experiment aimed at the molecular study of RASopathies, from the sequencing of the exome of a set of patients. The consortium seeks to investigate genetic aspects of RASopathies by using DNA sequencing technologies, providing support for the treatment of the disease. For this, a study that aims to process and analyze the sequences of patients diagnosed with RASopathies was established. This study comprises the use of a set of bioinformatics applications and, due to the significant amount of data to be processed, demands HPC.

A scientific workflow, called RASflow (https://github.com/mmondelli/rasflow; *Mondelli, 2018b*), was implemented to support the large-scale management of a bioinformatic experiment to analyze diseases associated with RASopathies and allows the identification of mutations in the exome of patients. Once the exomes of the collected patient samples are sequenced, the results are stored in a text file in the FASTQ format, which records the *reads* and their quality scores. The *reads* consist of sequence fragments generated by the sequencer. By using the results obtained through RASflow, a researcher can analyze and identify whether or not there is pathogenicity in the mutations present in the exome. RASflow was implemented in Swift, and the workflow model is shown in Fig. 8.

Activity 1 (version 2.1.0) receives a reference human genome and indexes it. The results are stored in files in the BT2 format, which records the indexes according to the sequence size. This activity requires large computational time, and its results are used in the analysis of each patient. Therefore, the workflow checks for the existence of BT2 files in the filesystem where it is executed. The activity is only performed when the files are not found.

Activity 2 receives the exome sequence of the patient in the FASTQ format as input, filters the reads with a certain quality, and maintains the result in the same format. The need to perform Activity 2 depends on the type of sequencer used to sequence the exome of the patient. If the Illumina sequence was used, for example, it is not necessary to perform this activity. If the sequencing comes from IonTorrent, the activity is performed in parallel.

Activity 3 receives the result of Activities 1 and 2 to perform the alignment of the exome and the reference genome sequence, resulting in a SAM format file and a log file with

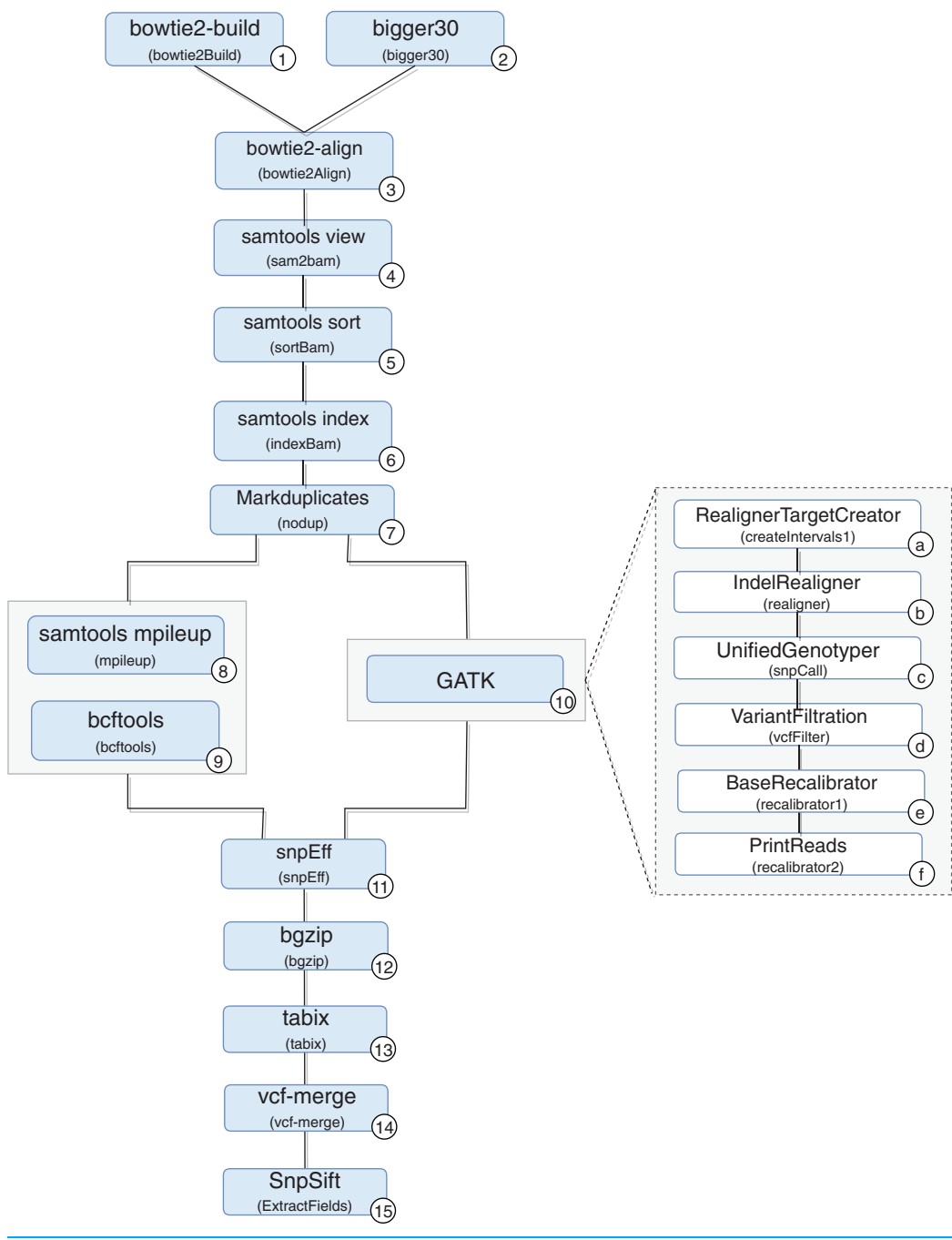

**Figure 8 RASopathy analysis workflow modeling.**

alignment statistics. The SAM file, the result of the alignment, is usually large and therefore difficult to analyze and manipulate. Activities 4–6 are used to compress, sort and index the SAM file, respectively, generating the BAM and BAI binary files that store the aligned sequences. The resulting files are used by Activity 7, which filters duplicate reads and produces a file in BAM format as well.

From this point on, mutations are identified by comparing the exome sequence of the patient and the reference genome. However, the workflow presents some variability,

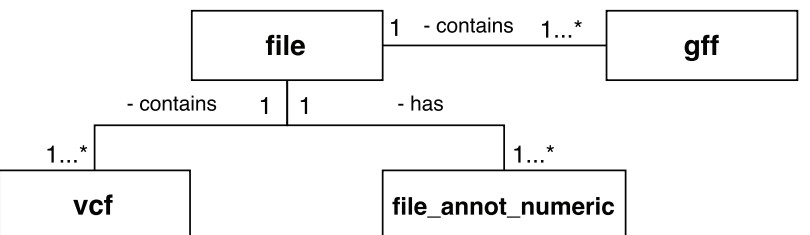

**Figure 9 Database entities for storing scientific domain annotations in RASflow.**

and the comparison can be made through two approaches: (i) using *samtools* (version 0.1.18) (*Li, 2011*) or (ii) the toolkit provided by *GATK* (version 3.4-46-gbc02625) (*McKenna et al., 2010*). A more detailed analysis of the differences between the two approaches that take into account biological aspects falls outside the scope of this work. However, it can be said that the GATK toolkit comprises more sophisticated techniques and the results obtained tend to be more accurate.

In the workflow variation that uses *samtools*, the BAM file produced by Activity 7 (version 1.100(1571)) and the reference genome are consumed by Activity 8, which performs the comparison. The result is used by Activity 9 (version 0.1.17), which converts the file from a binary format to the VCF format, which is responsible for storing the genetic variants found. The workflow variation using the *GATK* approach performs the set of Activities 10a–10f, indicated in the figure, which also produces a VCF file. The VCF file resulting from the execution of one of the variations is consumed by Activity 11 (version 3.4i) to predict the effects of the mutations found and to determine the pathogenicity, resulting in a set of VCFs. Note that Activity 11 is not a combination of the results of the samtools and GATK approaches. On the contrary, the activity processes the result of only one of the approaches, chosen by the user at the time of the workflow execution call. In this work, we also included Activities 12 and 13 (version 0.2.5) to compress and index the files resulting from Activity 11, respectively. The results are consumed by Activity 14 (version 0.1.13), responsible for combining this set of files. Finally, Activity 15 (version 4.2) applies a set of filtering activities based, for example, on the quality and quantity of occurrences of a given variant, generating a single final VCF file.

A set of scientific domain annotation extractors was developed for gathering and adding to the provenance database information from the following files: (i) the GFF file, which describes characteristics of the DNA, RNA, and protein sequences of the genome used as reference; (ii) the log file resulting from the execution of Activity 3; and (iii) the file in VCF format, which contains the variations found in the exome. The conceptual schema of the Swift database has been augmented to enable storage of this information. The following tables, as shown in Fig. 9, were added: *gff* and *vcf*, for storing the contents of the GFF and VCF files respectively; and the *file_annot_numeric* table, which records the contents of the log file in key-value format.

Next, we propose a set of queries to the RASflow provenance database to assist the scientist in the analysis of the results of the workflow. The queries were defined with one of

**Table 3 Alignment rate resulting from patients analysis in RASflow.**

| Patient | Alignment rate (%) |
| --- | --- |
| P1.log | 93.95 |
| P2.log | 94.52 |
| P3.log | 94.41 |
| P4.log | 94.48 |
| P5.log | 94.62 |
| P6.log | 94.58 |

**Table 4 Mutation list with the biotype and the name of the transcribed genes in the final VCF file of a patient.**

| Patient | Gene | Transcript | Name | Biotype |
| --- | --- | --- | --- | --- |
| P1.log | ENSE00001768193 | ENST00000341065 | SAMD11-001 | protein_coding |
| P1.log | ENSE00003734555 | ENST00000617307 | SAMD11-203 | protein_coding |
| P1.log | ENSE00003734555 | ENST00000618779 | SAMD11-206 | protein_coding |
| P1.log | ENSE00001864899 | ENST00000342066 | SAMD11-010 | protein_coding |
| P1.log | ENSE00003734555 | ENST00000622503 | SAMD11-208 | protein_coding |

the researchers responsible for the analysis of RASopathies. They are available for viewing in the BioWorkbench web interface corresponding to the RASflow experiment. That is, the scientist does not need to redefine them to obtain the results.

*Query 1*: "Retrieve the alignment rate of each patient's exome sequences relative to the reference genome." The alignment process allows identifying the similarity between the patient sequence and the sequence used as a reference. The result was obtained when the workflow executed Activity 2, which produced a log file that got parsed and added to the provenance database as domain data. This file records some statistics and among them the rate of alignment between the two sequences. This type of information is useful for the scientist, who is responsible for deciding what rate is sufficient for proceeding with the analysis. The alignment rate is stored in the *file_annot_numeric* table of the provenance database and can be retrieved through the query shown in Listing 3. Table 3 displays the result of the query, with the alignment rates for each of the patients used in the analysis of RASopathies of a given workflow execution.

**Listing 3 SQL query to retrieve the alignment rate of each patient's exome sequences.**

```
SELECT          file_id,  value
FROM            file_annot_numeric
NATURAL JOIN    staged_out
WHERE key LIKE  'overall alignment rate'  AND app_exec_id LIKE
                '%loss_all-run003-410029947%';
```

*Query 2*: "Retrieve the biotype and transcript gene name from the final VCF file." This type of result, without BioWorkbench support, was obtained through a manual scan of the GFF file, used as input in the execution of the workflow, and if the VCF file generated as the final result. In RASflow, these two files are imported into the *gff* and

*vcf* tables of the provenance database. To retrieve this information, one can use the query presented in Listing 4, which performs the join of the *vcf* and *gff* tables for a given workflow execution. The result is displayed in Table 4. Through the BioWorkbench interface for RASflow, the result is also presented as a table, but iteratively, allowing the scientist to filter the columns or search for names according to their needs.

**Listing 4** SQL query to retrieve the biotype and transcript gene name from the final VCF file of a patient.

```
SELECT        v.*,  g.Name AS nome,
              g.biotype AS biotipo
FROM          vcf v
NATURAL JOIN  file f
NATURAL JOIN  staged_out o
NATURAL JOIN  app_exec a
LEFT JOIN     gff g ON v.trid = g.ID
WHERE         a.script_run_id LIKE 'loss-run006-3737171381';
```

To evaluate the performance of the workflow, we used exome sequences of six patients as input, ranging in size from 8 to 11 GB. It is worth mentioning that a preliminary version of this workflow was developed in Python. In this work, we chose to develop it using Swift to take advantage of the parallelism offered by the system. By considering the variability of the workflow, executions were made for the two possible approaches: one using the *samtools* tool and the other using the *GATK* toolkit. In the approach using *samtools*, a reduction of ~77% of the execution time of the analyses was obtained, representing a gain of ~4 times in the execution with Swift compared to the sequential execution in Python. The approach using *GATK* had a reduction of ~80% in execution time, for a gain of ~5 times. The parallelism strategy of RASflow explored the simultaneous analysis of patients. In other words, each patient is analyzed sequentially and the parallelism is related to the number of patients used as input for the workflow execution. Thus, the total time of execution of the workflow is associated with the execution time of the most time-consuming patient analysis. Due to restrictions of the computational resource that we used, this performance analysis considers a single run for each approach using both Swift and the original Python version of the workflow.

Because it has been executed with a small set of patients, the workflow has a large run-time variability. However, genetic variant analysis can be applied to other diseases. For diseases such as cancer, for example, the volume of sequenced genomes is much larger and therefore, there is an opportunity to obtain higher performance gain through the approach used in RASflow. Also, we highlight the data used for the executions could not be made available because it is real patient data. However, we provide a sample of input data that can be used to run RASflow (*Mondelli, 2018e*).

## CONCLUSION AND FUTURE WORK

Large-scale bioinformatics experiments involve the execution of a flow of computational applications and demand HPC. The execution management of these experiments, as well as the analysis of results, requires a lot of effort by the scientist. In this work,

we demonstrate that the use of scientific workflow technologies coupled with provenance data analytics can support this management, allowing for the specification of experiments, parallel execution in HPC environments, and gathering provenance information.

To benefit from the use of scientific workflow technologies and support the whole process of scientific experimentation, we have developed the BioWorkbench framework. It was designed to use the Swift SWfMS for the specification and execution of bioinformatics experiments, and a web application for provenance data analytics. In this way, through BioWorkbench the user has access to a tool that integrates various features ranging from the high-performance execution of the workflow to profiling, prediction, and domain data analysis. We can observe that BioWorkbench enables a better scientific data management since the user does not have to directly manipulate the provenance database and the resulting files from the experiment execution. Another important aspect concerns the reproducibility of the experiment, which is facilitated by the provenance and the reproduction of the computational environment through a Docker container.

We used three case studies that model bioinformatic experiments as workflows: SwiftPhylo, SwiftGECKO, and RASflow. In addition, to the performance gains achieved by using Swift in BioWorkbench, we have demonstrated how the provenance allows the identification of bottlenecks and possible optimization opportunities in the execution of workflows. Also, we conclude that users can benefit from the application of machine learning techniques in provenance analysis to, for example, predict and classify workflow execution time. It is noteworthy that, during the development of this work, SwiftGECKO was integrated into the Bioinfo-Portal scientific portal (*Mondelli et al., 2016*). Through the portal, users can execute the workflow through a web interface in a transparent way, using geographically distributed computational resources managed by the Brazilian National System for HPC (SINAPAD).

### Funding
This work was partially supported by Brazilian funding agencies CNPq, CAPES, and FAPERJ. There was no additional external funding received for this study. The funders had no role in study design, data collection and analysis, decision to publish, or preparation of the manuscript.

### Grant Disclosures
The following grant information was disclosed by the authors:
Brazilian funding agencies CNPq, CAPES, and FAPERJ.

### Competing Interests
Ian Foster, Marta Mattoso and Daniel Katz are Academic Editors for PeerJ. Michael Wilde has an employment and ownership interest in the commercial firm Parallel Works Inc.

## Author Contributions

- Maria Luiza Mondelli conceived and designed the experiments, performed the experiments, analyzed the data, contributed reagents/materials/analysis tools, prepared figures and/or tables, authored or reviewed drafts of the paper, approved the final draft.
- Thiago Magalhães performed the experiments, analyzed the data, contributed reagents/materials/analysis tools, prepared figures and/or tables, authored or reviewed drafts of the paper, approved the final draft.
- Guilherme Loss performed the experiments, analyzed the data, contributed reagents/materials/analysis tools, authored or reviewed drafts of the paper, approved the final draft.
- Michael Wilde performed the experiments, analyzed the data, contributed reagents/materials/analysis tools, authored or reviewed drafts of the paper, approved the final draft.
- Ian Foster contributed reagents/materials/analysis tools, authored or reviewed drafts of the paper, approved the final draft.
- Marta Mattoso contributed reagents/materials/analysis tools, authored or reviewed drafts of the paper, approved the final draft.
- Daniel Katz contributed reagents/materials/analysis tools, authored or reviewed drafts of the paper, approved the final draft.
- Helio Barbosa performed the experiments, analyzed the data, contributed reagents/materials/analysis tools, prepared figures and/or tables, authored or reviewed drafts of the paper, approved the final draft.
- Ana Tereza R. de Vasconcelos conceived and designed the experiments, performed the experiments, analyzed the data, contributed reagents/materials/analysis tools, prepared figures and/or tables, authored or reviewed drafts of the paper, approved the final draft.
- Kary Ocaña conceived and designed the experiments, performed the experiments, analyzed the data, contributed reagents/materials/analysis tools, prepared figures and/or tables, authored or reviewed drafts of the paper, approved the final draft.
- Luiz M.R. Gadelha Jr conceived and designed the experiments, performed the experiments, analyzed the data, contributed reagents/materials/analysis tools, prepared figures and/or tables, authored or reviewed drafts of the paper, approved the final draft.

## Human Ethics

The following information was supplied relating to ethical approvals (i.e., approving body and any reference numbers):

This study was examined and approved by the Fernandes Figueira Institute Ethics Committee at Oswaldo Cruz Foundation, document number CAAE 52675616.0.0000.5269.

## Data Availability

BioWorkbench Docker container: https://hub.docker.com/r/malumondelli/bioworkbench/.

BioWorkbench source code: https://github.com/mmondelli/bioworkbench (DOI: 10.5281/zenodo.1243912).

SwiftPhylo workflow: https://github.com/mmondelli/swift-phylo
(DOI: 10.5281/zenodo.1241164).

SwiftGECKO workflow: https://github.com/mmondelli/swift-gecko
(DOI: 10.5281/zenodo.1241166).

RASflow workflow: https://github.com/mmondelli/rasflow (DOI: 10.5281/zenodo.1243176).

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
