# Peer review of "BioWorkbench: a high-performance framework for managing and analyzing bioinformatics experiments"

_PeerJ, doi:10.7717/peerj.5551_

## Round 0.1 · original submission · Major Revisions

I agree with the reviewers, it would be useful for the authors to revise the manuscript and respond to the reviewers comments appropriately

·

Basic reporting

The article has clear and unambiguous language, explaining well the methods and background of the work.

The coverage of related scientific workflow systems, while extensive, seem a bit dated or HPC-centric; not including popular bioinformatics workflow systems like Nextflow, Common Workflow Language, Toil or Galaxy. Relevant reviews could be https://doi.org/10.1093/bib/bbw020 and https://doi.org/10.1007/s41019-017-0047-z

The article fails to describe clearly what it the new contribution to the field, as it suggests a BioWorkbench "framework", but it is unclear if this means a general methodology (which should then be further explained as such) or a piece of software (which should then be made available).

Buried in the text is potentially interesting research on using machine learning for predicting execution times and analysing provenance, but as this is not the focus of the article (submitted as "Bioinformatics Software"), this gets a bit in the way of introducing the BioWorkbench framework. I would welcome a change of focus (perhaps separate paper) exploring this approach in detail, but then it would need to relate to other research on predicting computational parameters.

My main challenge with the paper is that it lacks focus and covers several aspects, seemingly proposing new bioinformatics software which is only partially made available.

Experimental design

The authors propose reusing the Swift language for writing bioinformatics workflows because of its scalability benefits and support for high-performance compute backends.

The paper explains in detail the layers of the proposed BioWorkbench architecture which extends Swift to analyze the provenance using R scripts and a web UI. The BioWorkbench workflows reuse existing bioinformatics tools like picard and bowtie, some of which are wrapped in Python scripts. The scripts are then composed within Swift which provides parallelism between the chained tool executions.


The internal provenance model of Swift is explored and exercised with several example queries, however this has not been related to existing approaches to standardizing workflow provenance, such as PROV, OPMW, ProvOne, D-PROV and Research Object. Extensive literature on workflow provenance exists, see for instance the recent IPAW conference proceedings.

It is not well justified why BioWorkbench should use a custom Swift-specific provenance model rather than one of the existing models, other than that this is what was already given by the Swift.


The article uses three bioinformatics pipelines as use-cases, and explains well how these were implemented with the Swift language. Two of the workflows were provided as GitHub links
https://github.com/mmondelli/swift-phylo and https://github.com/mmondelli/swift-gecko - while perhaps the most complete workflow Rawflow is only provided as a manually composed Docker image. I inspected this to find the third source reference https://github.com/mmondelli/rasflow

All of these URLs are subject to change in availability and content; I would request the authors to use the Zenodo-GitHub integration
https://guides.github.com/activities/citable-code/ to make Software DOI citations with permanent availability.

For the Docker image perhaps a Zenodo upload of the `docker export` tar, as well as a DOI citation also for (apparently) the underlying Dockerfile recipe https://github.com/mmondelli/docker-masters/ (I had to browse GitHub to find this)

It is not specified which Swift version is used (citation Wilde2011) - the above Dockerfile hints of http://swift-lang.org/packages/swift-0.96.2.tar.gz (latest version from 2015). Similarly explicitly declaring the version of each tool used is generally important for bioinformatics pipelines - the BioWorkbench does not seem to handle installation of bioinformatics software (e.g. from BioConda) and these are instead added manually to the Docker image or directly as binaries in the GitHub source code repository, but missing their attribution and open source licenses.

The authors are asked to improve their BioWorkbench distributions to comply with the open source licenses of the tools and software they redistribute.


I will commend the authors on providing a Docker image for reproducibility, however this seems to be incomplete and lack instructions for usage. Some care seems to be needed by users, setting up paths and shell environment variables for running the workflows, but this process is not well explained. The image was still valuable for me in reviewing this paper, but I believe in its current form it is not reusable as "Bioinformatics Software", and so I cannot recognise this as the proposed "BioWorkbench framework".


I am generally unable to replicate any of the results of the paper, even when the authors have a provided Docker image. The main reason for this is that the image lacks documentation, and is incompletely setup - e.g. I had to perform several manual installation steps in the shell of the docker image, but was still unable to generate provenance information. For instance the workflow seems to require a postgres database, which is neither set up or correctly configured from the workflow.

Some of the workflows contain brief documentation, and I managed to run those inside the Docker image. I did not assess their execution beyond the fact that they produced bioinformatics-like result files.


I must admit I am unable to locate the described "BioWorkbench" software, neither on GitHub or (to my knowledge) in the Docker image, so I am unable to verify any of the claims on the provenance browsing or web interface.

The paper describes workflow execution with bacterial genome sequences, but these datasets are not cited or otherwise made available.

The Weka configurations, customization source code, source datasets and result datasets have not been made available, and are not present in the Docker image. The authors do not specify which version of Weka was used. I am unable to verify any of the claims around machine learning.

Validity of the findings

I have not reviewed the validity of the bioinformatics pipelines or the machine learning approach.

Some of the data are only made available as graphical figures (e.g. Figure 5, 6), and others as tables or inline in text. If this paper was to focus on benchmarking or machine learning approach, the actual results data should also be made available, e.g. on Figshare or Zenodo.

The paper discusses speed-ups in parallelism of the Swift workflows compared to sequential execution. As this would be the case for most "embarrassingly parallel" pipelines in bioinformatics, commonly using distributed and scalable grid/cloud workflow systems, I don't think there is much information gain in providing the % speed-up here, at least without comparisons to other existing approaches (even if this would just show a comparable scaling factor).

Additional comments

Hi, I am Stian Soiland-Reyes http://orcid.org/0000-0001-9842-9718 and believe in open reviews.

The complete review is also available at the (secret) URL
https://gist.github.com/stain/eedf7c3a355e347863a2381a2f55e957
for easier reading and more hyperlinks.

This review is licensed under a
Creative Commons Attribution 4.0 International License
http://creativecommons.org/licenses/by/4.0/


This paper seem to show an area of interesting research by the authors, but tries to cover a bit too many things at once.

Covering large aspects of a piece of software might be appropriate once it is in a mature, reusable state or have an established user-base - but as this stands I get more of the impression that the proposed BioWorkbench is mainly a snapshot of one person's research methodology.

In the guidelines for PeerJ submissions of Bioinformatics Software it is strongly recommended that the software is made available open source -- this is so that it can be reused and expanded by other researchers. The proposed BioWorkbench is either incomplete or insufficiently described and packaged. Further work would be to mature the software so it is a tangible piece in its own right (e.g. with its own GitHub repository and reproducible installation instructions), rather than just a usage pattern of the Swift language.

Achieving scalability with distributed and parallelized pipelines is common practice in bioinformatics, and while http://s.apache.org/existing-workflow-systems currently lists 200 workflow systems (including Swift), there is of course room for BioWorkbench as well, particularly with focus on provenance and predicting computational parameters with machine learning.

Where this paper goes interesting is in the machine learning approach, in particular in relating computational parameters with the domain-specific input parameters like sequence length -- this is in particular a challenging area for metagenomics when trying to predict good job allocations in distributed computing (e.g. required memory is not known in advance). As this is just one of the many areas of this paper, this does not get the space it deserves. This is also the part of the research where all data should be easy to share and the post-workflow provenance analysis results easy to reproduce, had they been made available as datasets together with this paper - sadly here most details were missing.

I would recommend the authors to:

* Split out machine learning approach to a new paper. A citation to a first draft at arxiv can be used as placeholder here.
* Improve BioWorkbench as a separate piece of software (git repo, install instructions)
* Make sure the software and workflows can be re-run (and hopefully re-used) by other people
* Make sure the Docker image is complete and runnable out of the box. Include detailed step-by-step instructions for each workflow as well as for how to do post-run provenance analysis.
* Relate or convert custom provenance to existing PROV standards. This could for instance be an export script to one of the PROV formats.
* Refocus this paper to have a single focus -- this might be BioWorkbench provenance, the bioinformatics workflows, or machine learning.
* The paper needs to give a stronger claim to what is the proposed contribution, and add a more thorough Discussion section that can then compare this to other existing work. This might require updates to the Background section.
* Ensure all contributed software and datasets are provided with the article, preserved (and given DOI) using services like Zenodo or FigShare; in addition to the GitHub/DockerHub approach.

Reviewer 2 ·

Basic reporting

Overall the article is well written and easy to follow.

Experimental design

Overall the approach is sound, and the proposed architecture shows real benefits.
The remaining questions (see detailed comments below) can be addressed in a minor revision.

Validity of the findings

The findings are very plausible. But some extra care should be taken to address questions raised in the detailed comments below.

Additional comments

* INTRODUCTION
- P2, L50: life-cycle of a scientific workflow: design, execution, and analysis
=> What is meant here by 'analysis'?
Is it the analysis of the workflow outputs/results, or the analysis of the workflow itself?
It seems to be the former. Maybe simply call it "result analysis" to avoid a possible confusion?

- P2, L72: You refer to BWB as a 'framework' but also as a 'tool'.
Please elaborate and clarify what it is: it could be both, but there are different expectations associated with those names. For example, if BWB is a tool, then is it something I can download, install, and run in my own environment (at my institution)? Or is it a service/webapp, and if so, who can use it and how? Or is it framework that you prototyped and that you're reporting on, but there is no expectations for other to be able to use it?
(Later in the paper, this becomes more clear, but I suggest to use consistent wording / positioning of BWB throughout the paper.)

- P2, L77: A reduction of 98.9% sounds really good, but the maths seems unnecessarily complicated.
Why not just say something along the lines of "in the case of SwiftPhylo we achieved a speedup of about 100x, i.e., a 13.35 hours execution time was reduced to 8 minutes"

* RELATED WORK

- P2, L88: ".. Pegasus .. enables the specification of workflows through the XML format."
I don't think this comment is particularly helpful as it seems to involve a kind of category error:
Is it really worth mentioning what exchange *syntax* is used for a workflow (XML, JSON, YAML, ...)?
What about the specific XML (JSON, YAML) schema being used?
In fact, the schema itself is also one level of indirection away from understanding a more important aspect of the system, i.e., the underlying model of execution employed by Pegasus:
Is the Pegasus model similar (or identical?) to the model of Condor/DAGMAN? Does Pegasus use partially-ordered task dependency graphs (DAGs), where a tasks can only start after all its upstream dependencies have completed?
Or does Pegasus support other execution models, e.g., pipeline parallelism (maybe with promises and futures)?

- P2, L90: ".. provenance .. can be queried using the SPARQL language".
Similarly to the point above, the use of SPARQL seems somewhat secondary. What is the provenance model that is being employed? What *kind* of questions are supported by the underlying provenance model?
For example, you advertise that in BWB you have two kinds of provenance that can be employed, i.e., monitoring and improving performance, and to query domain data dependencies. Is this also the case for Wings/Pegasus? If not, what is the nature of the provenance queries they support?

- P3, L99-103: In contrast to some of the "formal/syntactic differences" mentioned above, the following are examples of the kind of more meaningful differences that I think readers will appreciate when combing through the Related Work section:
-- "[SciCumulus] .. is distinct in that it allows provenance queries .. during the execution .."
-- ".. we use the Swift SWfMS because it transparently supports the execution of workflows in different HPC environments"
-- " .. it supports workflow patterns such as conditional execution and iterations that not supported by .. "

- P3, L123-126: "We show that it is possible to combine provenance and domain data for more detailed analysis. Also, we demonstrate that these analyses can benefit from machine learning techniques for extracting relevant information and predicting computational resource consumption that are not readily detected by queries and statistics from the provenance database."

These findings and results seem to be buried here in the wrong (Related Work) section. They should be moved (or restated) prominently elsewhere in the paper, e.g., under "Contributions" or "Results".

- The Related Work section could also benefit from a discussion of/comparison with the following:

-- Kanwal, Sehrish, Farah Zaib Khan, Andrew Lonie, and Richard O. Sinnott. 2017. “Investigating Reproducibility and Tracking Provenance – A Genomic Workflow Case Study.” BMC Bioinformatics 18.

-- Altintas, Ilkay, Jianwu Wang, Daniel Crawl, and Weizhong Li. "Challenges and approaches for distributed workflow-driven analysis of large-scale biological data: vision paper." In Proceedings of the 2012 Joint EDBT/ICDT Workshops, ACM, 2012.

* MATERIALS and METHODS

- SPECIFICATION AND EXECUTION LAYER
-- Nice introduction to Swift. However, I wonder how one should think of Swift as a model of execution? Is it elements of a functional scripting language. But there is also this (P4, L160):
"All expressions in a Swift script whose data dependencies are met are evaluated in parallel. "
This seems to indicate that Swift also has dependency driven features similar to MAKE or Condor/DAGMAN.
Can you clarify/elaborate a bit?
(later in the paper you talk about "activity chaining" .. so maybe mention it earlier or make a forward reference?)

- P4, L158: something missing here: ".. applications outside Swift should be executed, their inputs and outputs."

- P4,L175: something missing here: ".. analyzing the data derivation process of the experiment possible."

- Figure 2 is useful to get an idea of the provenance model. However, I noticed a symmetry w.r.t. the cardinality constraints of stage_in and stage_out, where I would have expected an asymmetry: It seems a file can be consumed by many app_execs but should probably only be produced by a single one (to avoid a write conflict) - or am I missing something? Please elaborate!

- While the model in Figure 2 is nice and simple, there must be a lot more going on in terms of the "complete provenance model".
You emphasize essentially two kinds of uses of provenance information, i.e., for monitoring (and then improving) execution provenance, and for querying science data.
Can you say a bit more about how the "dependency model" in Figure 2 is connected with other provenance information and metadata needed to answer the two different kinds of provenance use cases?

- P6: You describe the use of Weka and ML techniques in support of provenance analysis.
This section is a bit disconnected from the rest of the story.
And while I understand from the rest of the paper that BWB is essentially a framework and "demo"/proof of concept (not meant as a production tool), I still wonder what the challenges were in bringing together so many different system components: Swift, R, Weka, ...
For example, would you anticipate a production-level version of BWB to also use that many different systems?
What are the repercussions in terms of system maintenance, with all constituent tools evolving independently?
This might be worth mentioning or discussing..

* RESULTS AND DISCUSSION

- P6, L268-270: "It is worth mentioning that the computational resource is shared with other users and was not dedicated to these executions. This can be considered as one of the factors influencing the performance gains. "

First and foremost, I commend you for including this frank comment!
However, it raises and important question: the 100x speedup reported elsewhere in the paper: how "real" or robust is it?
If some of it might be attributable to differences in system load, it seems something has to be done about it. There seem to be a number of options, for example:
-- Couldn't you use some of your provenance data, especially on execution monitoring, to do a careful accounting of total cycles spent?
(in that case, it would be interesting to see whether/how cpu-time, user-time, wall-clock time etc. are measured and accumulated)
-- Or what about repeating the experiment many times, to see what the variation across runs (and supposedly different load situations) is?

I understand that meaningful benchmarking is a non-trivial exercise all by itself, but it seems that a bit more could be done here to sort this out. After all, provenance collection and reproducibility are a major theme of the paper, and it would be ironic to not be able to reproduce your findings.

For example, how about re-executing some of the experiments, maybe at a smaller scale, on a dedicated system to see what happens with the 100x speedup? Since you have a docker container for the system/demo available, wouldn't it be easy to rerun some experiments?
But I fear that in practice things are quite a bit more complex, not just because of the limited availability of hardware (say the 160 cores, 2TB RAM machine; again maybe a smaller machine would do for a replication study), but also because of having a docker container probably isn't quite the silver bullet for computational reproducibility as we're often led to believe.

If it is practically feasible to do a little replication experiment to see whether you can get a similar 100x speedup, then I encourage you to do so and report on the findings. On the other hand, if that's not practically feasible, I suggest to make clear that the 98% reduction / 100x speedup has to be taken with a grain of salt. After all, based on this paper someone else might try and replicate your findings and might have a hard time doing so, despite all the right elements being brought into play (use of detailed provenance information, docker containers etc.)

In particular, I think this sentence needs to be reworded:
"Through the provenance data, we show that the framework is scalable and achieves high performance, reducing up to 98% of the case studies execution time."

While the "up to" makes it clear that one cannot expect this speedup in all case studies, it doesn't rule out that for the particular case study (SwiftPhylo) this result is replicable (incidentally, you report 98.9% elsewehre in the paper, which is closer to 99% or the 100x speedup).
On the other hand, Listing 1 (SwiftPhylo spec.) seems to exhibit an "embarassingly parallel" use case, in which such a speedup isn't all that surprising (especially on corresponding hardware). However it is not clear why one would need provenance data to establish this speedup.

- P6, L272: "The Docker container was built for reuse purposes only, to encapsulate the components that compose the framework."
This begs the question: where any of the use cases run with the Docker container at all? If not, why not?
How would one know that the container "works" if it isn't tested with (at least downscaled versions of) the use cases?

- P7, L277: "The construction of phylogenetic trees allows understanding the evolutionary relationship between organisms, determining the ancestral relationships between known species"
I suggest to reformulate since the output of this analysis is a hypothesis (or adds some evidence to an existing hypothesis). The resulting tree doesn't imply that an actual relationship has been determined with certainty (although it might increase the confidence that certain relationships exist). Maybe something along the lines of ".. allows to study the evolutionary relationships between organisms.." could be used to clarify that the nature of a result of a phylogenetic analysis isn't as clear cut as, e.g., the prime factorization of a large integer.

- P7, L281 SwiftPhylo also uses Perl (adding to the already complex software dependencies).
You mention elsewhere in that paper that you make a container available.
Do you also share the docker recipe? It would be illustrative to include it among the supplementary material of this article to give the reader a full appreciation of the "rich" dependency structure and complexity of putting this application together.

- Figures 5 (in particular) and 6 increase the confidence of the reader that your near 100x speedup for SwiftPhylo on the appropriate hardware is "real" (again, Listing 1 seems to hint at an "embarassingly parallel" situation, which is nice to have indeed).
However, it still would be helpful to have an understanding how large the (missing!) error bars would be.
It seems that even with only 10 cores, you get in the range of (wall-clock?) runtime of about an hour.
Have you rerun the experiments a few times and are those runtimes available in your provenance store?

- P10, L 345-347: "The execution time decreased from ∼2.1 hours to ∼6 minutes, representing a reduction of ∼96.6%. Also, the workflow was ∼30 times faster when executed in parallel, using 160 processors."
This makes it sound as if the 30x speedup is a measurement separate from the 96.6% reduction, but both refer to the same reduction from 2.1 hours to 6 minutes, correct?
Please reformulate.
For example, instead of "Also ..."
rather something like : ".. this corresponds to a roughly 30x speedup"
Incidentally this seems to be in terms of wall-clock time (it can't be cpu-time, right? After all, the total cpu-time would be approximately the same, when costing is based on total time spent across all cores..)

- P10, Table 1: Just confirming - is the table sorted in "workflow order"? (might be worth mentioning)
Or how about sorting it by duration (descending) to make it easier on the reader?

- P11, Equation (1): Can you explain this equation a bit more? It's a bit hard to parse..
Is this an arithmetic equation? Then what is the meaning of "v" (looks like a logical disjunction)
After spending a bit of time with it, I think I understand it now, but still encourage to try and make it clearer.
For example, how about something like this:

- 931.4025 * (word_length \in {10,12})
- 2704.5534 * (word_length = 10)
+ 0 * total_written
- 1006.3049 * (total_genomes \in {20,30,40})
- 547.6854 * (total_genomes \in {30,40})
... etc

When computing this regression, how many data points did you have?
From how many different runs?

- P12, L384: ".. these experiments demonstrate that is possible to generate simple models able to efficiently predict the computational time demanded by the workflow execution, both in terms of continuous or discrete times. "

Is this really demonstrated by these experiments? To test a prediction, wouldn't you have to extrapolate (or interpolate) to a scenario that you haven't yet measured, and then make the measurement / benchmark to see how close your prediction was? If I understand your analysis right, you didn't do that. Instead you had a set of data points (again: how many? Across how many runs? How many of those were "replications"?) and you fit e.g., a regression model to it. Is that correct? In that case, I suggest you reword the statement on "demonstrating... to efficiently predict..."

What I think your experiments show nicely though is that you can get some insight into the performance bottlenecks and better understand the execution behavior.

- P13, L405: ".. between *the* Bioinformatics Laboratory .." (insert article)

- P13, L410: "... a study that counts on the processing of the sequences ... "
Not very clear what "counts on" means here. Does the study make use of the sequences ?

- P13, L420: "Therefore, the workflow checks for the existence of BT2 files in the filesystem where it is executed."
This sounds like active polling? (repeated checking whether the previous step finished)
How often is this check performed? Are there trade-offs and possible race conditions?

- P14: Can you clarify whether the merge at Step 11 in Figure 8 is such that *either* the samtools branch *or* the GATK output is processed? That is, there is no comparison or merge of the results from the left and right branch?
If that's the case, could a comparison be used, e.g., to compare and double-check the quality of the different branches?

- Figure 9 has some of the additional information I was curious about earlier (so maybe make a forward reference?)

- P15, Listings 3, 4: Although a bit cryptic, it is nice to see that user/scientist's questions become executable SQL queries. Together with the narrative for "Query 1" and "Query 2", the reader gets a good idea how this works.

- P16: Just like above: For the various speedups reported (e.g. 4x, 5x), how many times did you run these experiments? If it was only once, it should be mentioned. If it was many runs, why not report the averages (which I assume these numbers are) and the standard deviation?

---

## Round 0.2 · Minor Revisions

I agree with the reviewer, that the manuscript would benefit from the minor changes suggested.

[# Staff Note: Reviewer 1 declared a potential Conflict of Interest, and the Editor was aware of this when making their decision #]

·

Basic reporting

The text is well-written and has a good structure.

I think the code citations should be moved to References instead of footnotes (they are now first class citable), but I understand if the GitHub hyperlinks should remain as footnotes so that the PeerJ typesetters can add them as inline hyperlinks.

I have attached some Detailed Comments line by line. In summary:

* Described usage is not "HPC" -> use "HTC"
* Fixes in Taverna details
* Fixes in Nextflow details
* How are required tool binaries installed on compute nodes?
* Shiny library vs HPFS-Prof vs new "workbench" code?
* Need deeper NCBI links
* Some grammar and phrasing suggestions

Experimental design

While the Docker container containing the Workflow script is great for reproducibility (as I partially verified under Validity), it seems hard for workflow development. In the way this is described, BioWorkbench is not a framework for writing any scientific workflow, but a framework for capturing a particular workflow, e.g. the provided Docker container captures 3 specific workflows and the Dockerfile is similarly specialized to add their dependencies.

As such I think you can say you are providing a Methodology rather than a Framework. If I am to create my own BioWorkbench that does a slightly different bioinformatics workflow I would need to make my own Dockerfile with my particular dependencies, create a brand new Swift script using the same patterns, build a new Docker image and test it inside the Docker container (or use vim inside the container to temporarily fix something).

Such a development methodology is however not mentioned in the paper, and it can be confusing to readers that won't know if BioWorkbench is just something you did (and we can learn from it and play with your demo), something we can repurpose (changing workflows, etc), or something general that can be reused (it can consume any workflow). I think now the truth is somewhere in the middle.

The text should more clearly make the point what maturity level BioWorkbench is at, and show what is the actual contribution. Now the text still gives the impression that BioWorkbench is presented for users, but I would argue that it is this methodology that is the contribution at this stage. I would therefore still be a bit careful about calling it a "framework", perhaps "prototype framework"?

Validity of the findings

Thanks to the updated README I was able to successfully build the Docker image from the bioworkbench recipe. The build takes a while to complete as various R packages are compiled.

I was unable to retrieve the (now automatically built) image from the Docker hub on my laptop, as I ran out of diskspace when writing swift_provenance.db.

I was also unable to retrieve the hub image on a spare server I had available (2x2 TB disk), due to the larger image size (15 GB) and that I was using the Docker LVM production configuration, which had a default max of 10 GB. Setting dm.basesize should improve this; unfortunately I did not have time to reconfigure my server.

This is unlikely to affect desktop Docker users which don't seem to hit this limit, but the authors are still advised to try to split into two Docker images, one smaller base image (tools, BioWorkbench, SwiftProf); and an extension image (using Dockerfile FROM that adds the larger pre-populated database.

Within the shell of the Docker image I was able to run the embedded swift-gecko and swift-phylo workflows according to their README files. As in my previous review I have not assesed the bioinformatics validity of these workflows, but confirm that they produced expected kind of outputs.

While intermediate files are kept separate in directories like run001, the workflow outputs are added directly to the working directory, which makes it hard to distinguish inputs, workflow, binaries and outputs. This may be improved by restructuring the workflow directories or workflow script.

As not enough reference data (e.g. for SwiftGECKO) was included, I was unable to verify any of the % speedup claims, both workflows finished rather quickly, but I did not find any obvious way to modify the number of concurrent jobs when testing on a multi-core server.

I was NOT able to run the embedded rasflow workflow as it did not contain any example sequences. Attempting to read the Swift workflow "loss_all.swift" is quite confusing, as it is a single flat linear script file that mixes shim scripting, variable declarations, control calls like snpCall() and actual code calls. For instance, the 6-step GATK sub-workflow of Figure 8 is 20 lines in the middle of a single large Swift file. The step names in the Figure 8 do not match the code, I would have expected the Figure steps to appear as functions or so. The authors are recommended to refactor the RasFlow workflow script for greater readability.

The authors argued in the rebuttal that RasFlow inputs are missing due to patient confidentiality. I think readers who try BioWorkbench would be grateful if some smaller example sequences and GFF/GTF files could be added instead, so that the RasFlow workflow also can be tested out of the box, even if it would not be able to reproduce the particular provenance data included.

I was able to access the embedded HPSW-Prof profiling web server. The user interface is somewhat confusing as there are no instructions (what goes into "Script Name"?), but I was eventually able to lower the "Date Range" to 2015 to see older runs and - after a wait - access the embedded provenance traces of executions of RASFlow "loss_all.swift". This showed figures similar to the paper's Figure 3 and Figure 6.

Unlike the Figure 6 the UI's "Level of Parallelism" for RASFlow seems to indicate only 4 concurrent executions (the figure is for SwiftPhylo). The number of cores or configured concurrent activities are not shown, but the table gives detailed execution stats per task of the workflow, e.g. "avg_duration" (unit is seconds?), percent CPU used, file system reads (unit is bytes?), and other parameters like "avg_max_rss", "sys_percent" and "user_percent" that remain unexplained.

As these are features/critizism of the previously published (and cited) HPSW-Prot this review does not go further in detail, but note that the consistent colour scheme is helpful to cross-reference tables and graphs.

The included provenance database do demonstrate the Domain Info added to the database after parsing , which includes 94,651 mutations from the 6 patients (anonymized as S1, S2, etc). I have not assessed if making these mutations public are compatible with patient confidentially, as no Ethical Approval assessment or similar data management plan for the RASOpathies study was cited.

The provenance database did not include data from the SwiftPhylo workflow, which was used for the execution speedup discussion in the paper, so I tried to add it:

swiftlog -import-provenance run001

as referenced in swift-gecko README. As detailed in attached, this did however fail for both swift-gecko and swift-phylo with a ValueError.

As mentioned above I was also unable to run RasFlow, and therefore unable to import any fresh workflow runs into the provenance database to visualize them in the web interface.

While the authors have taken significant steps to improve the packaging and documentation of the software, I am still under the impression that this is more of a demonstration snapshot that illustrate a methodology of developing and inspecting bioinformatics workflows with Swift Language and the previously published HPSW-Prof, rather than a General Availability framework (software or workflow package) for bioinformatics researchers.

From the Docker image I was able to run the new Weka instructions; after setting CLASSPATH to the weka.jar; the weka GUI does not work inside Docker. I got the same linear agression algorithm constants as in the paper's listing (1). As Weka output is quite verbose and outside my expertise I did not verify the remaining Weka calculations, but confirmed that they all executed well.

The rebuttal explains that Swift provenance model predates OPM and its successor W3C PROV, and mentions a swift_provenance-to-opm.sh script to generate OPM, which I tested; however this is not mentioned in the manuscript at all, which I find odd given that several pages are about provenance. It can be listed as future work to update this script to generate PROV instead of OPM.

Additional comments

Hi, I am Stian Soiland-Reyes https://orcid.org/0000-0001-9842-9718 and believe in open reviews.

The complete review is also available at the (secret) URL https://gist.github.com/stain/eedf7c3a355e347863a2381a2f55e957 for easier reading and more hyperlinks.

This review is licensed under a Creative Commons Attribution 4.0 International License http://creativecommons.org/licenses/by/4.0/

This revision has improved significantly on most of my previous concerns. I am particularly pleased with the new completeDocker images, improved instructions and Zenodo code citations. Software used by the workflows (e.g. samtools) is now rigourisly versioned and cited, this is also a good improvement.

There are still a couple of issues with the software, I was unable to actually import the provenance to the database, which would seem central to the argument of the paper. This seems to be a bug in the log parsing, perhaps some sensitivity to how the workflow is invoked. Following the README step by step within the Docker image should work.

I was also not able to run the RASFlow workflow as no examples are provided, yet this is the prime workflow of the paper and the subject of the machine learning approach.

While the manuscript now makes it clearer that the Docker image is for demonstration purposes, I think it can still come across like if BioWorkbench is a piece of software to be used by bioinformaticians. I think the manuscript can be honest about the current maturity level of the prototype framework, and rather present the main contribution as a methodology for building and retro-inspect paralellizable workflows using Swift and provenance database.

See https://gist.github.com/stain/eedf7c3a355e347863a2381a2f55e957 for details.

---

## Round 0.3 · accepted · Accept

Thank you for your revised version.

#